# Vertical structural complexity of plant communities represents the combined effects of resource acquisition and environmental stress on the Tibetan Plateau

Changjin Cheng[1,2], Jiahui Zhang [3] ✉, Mingxu Li [4], Congcong Liu[4], Li Xu[4] & Nianpeng He [3,5] ✉

The vertical structural complexity (VSC) of plant communities reflects the occupancy of spatial niches and is closely related to resource utilization and environmental adaptation. However, understanding the large-scale spatial pattern of VSC and its underlying mechanisms remains limited. Here, we systematically investigate 2013 plant communities through grid sampling on the Tibetan Plateau. VSC is quantified as the maximum plant height within a plot (Height-max), coefficient of variation of plant height (Height-var), and Shannon evenness of plant height (Height-even). Precipitation dominates the spatial variation in VSC in forests and shrublands, supporting the classic physiological tolerance hypothesis. In contrast, for alpine meadows, steppes, and desert grasslands in extreme environments, non-resource limiting factors (e.g., wide diurnal temperature ranges and strong winds) dominate VSC variation. Generally, with the shifting of climate from favorable to extreme, the effect of resource availability gradually decreases, but the effect of non-resource limiting factors gradually increases, and that the physiological tolerance hypothesis only applicable in favorable conditions. With the help of machine learning models, maps of VSC at 1-km resolution are produced for the Tibetan Plateau. Our findings and maps of VSC provide insights into macroecological studies, especially for adaptation mechanisms and model optimization.

Vertical structural complexity (VSC) in plant heights, which is physical niche partitioning in above-ground space, is thought to be an important property for a specific plant community[1]. VSC is closely linked to various ecological processes[1,2]. For example, in research on the diversity-productivity relationships, the classical complementary effect[3] proposes that the spatial complementarity due to VSC is a key determinant of overyielding in species-rich communities[4–6], as more complex vertical structures mean stronger spatial niche partitioning, thereby reducing competition[7,8] and leading to the unique occupancy of niche axes such as

light[2,4]. However, few large-scale studies provide direct evidence that VSC links the diversity-productivity relationship because VSC is rarely quantified. We have conducted a statistical analysis of 136 documents from various locations worldwide, and only 10 of them have quantified VSC (Table S1). In addition, a recent study showed that VSC has a stronger ability to explain spatial variation in productivity than species diversity at regional scales[9]. Hence, it is imperative to investigate the underlying mechanisms of VSC spatial variation and develop high-resolution VSC atlases, which will aid in understanding the diversity-

[1]State Key Laboratory of Plant Diversity and Specialty Crops, South China Botanical Garden, Chinese Academy of Sciences, Guangzhou 510650, China. [2]School of Ecology and Nature Conservation, Beijing Forestry University, Beijing 100083, China. [3]Key Laboratory of Sustainable Forest Ecosystem Management - Ministry of Education, Northeast Forestry University, Harbin 150040, China. [4]Key Laboratory of Ecosystem Network Observation and Modeling, Institute of Geographic Sciences and Natural Resources Research, Chinese Academy of Sciences, Beijing 100101, China. [5]Northeast Asia Ecosystem Carbon Sink Research Center, Northeast Forestry University, Harbin 150040, China. ✉e-mail: zhangjiahui@nefu.edu.cn; henp@igsnrr.ac.cn

productivity relationship and accurately predicting spatial variations in productivity.

The knowledge of the spatial variation of VSC at large scale is mainly explained by the physiological tolerance hypothesis (PTH)[10–12]. PHT reveals VSC spatial variation mainly from the perspective of regional differences in resource availability[9]. The PTH proposes that more adequate resources (e.g., more humid and warmer climates) support greater plant height, more complex species composition, and a wider spectrum of plant functional strategies (for example, greater crown plasticity and more shade-tolerant species), resulting in greater VSC[10,11]. Differently, related studies at the local community scale have more often attributed the complex vertical structure to asymmetric competition between plants. Competition often results in

inconsistent individual sizes and promotes complex vertical structures, because larger individuals gain more resources per unit of biomass and should inhibit the growth of their smaller neighbors[13]. In addition, the intensity of competition between species will be greater in areas with superior environments[14].

The resource acquisition strategies (resource availability and competition for resources) highlighted by PTH and asymmetric competition help to some extent the understanding of VSC spatial variation[10,15]. However, a trade-off exists among the plant species in communities to acquire resources for rapid growth in "favorable conditions" (when resource availability is high and environmental stresses are low) *vs.* its ability to acquire higher fitness to avoid mortality in "extreme conditions" (when resource

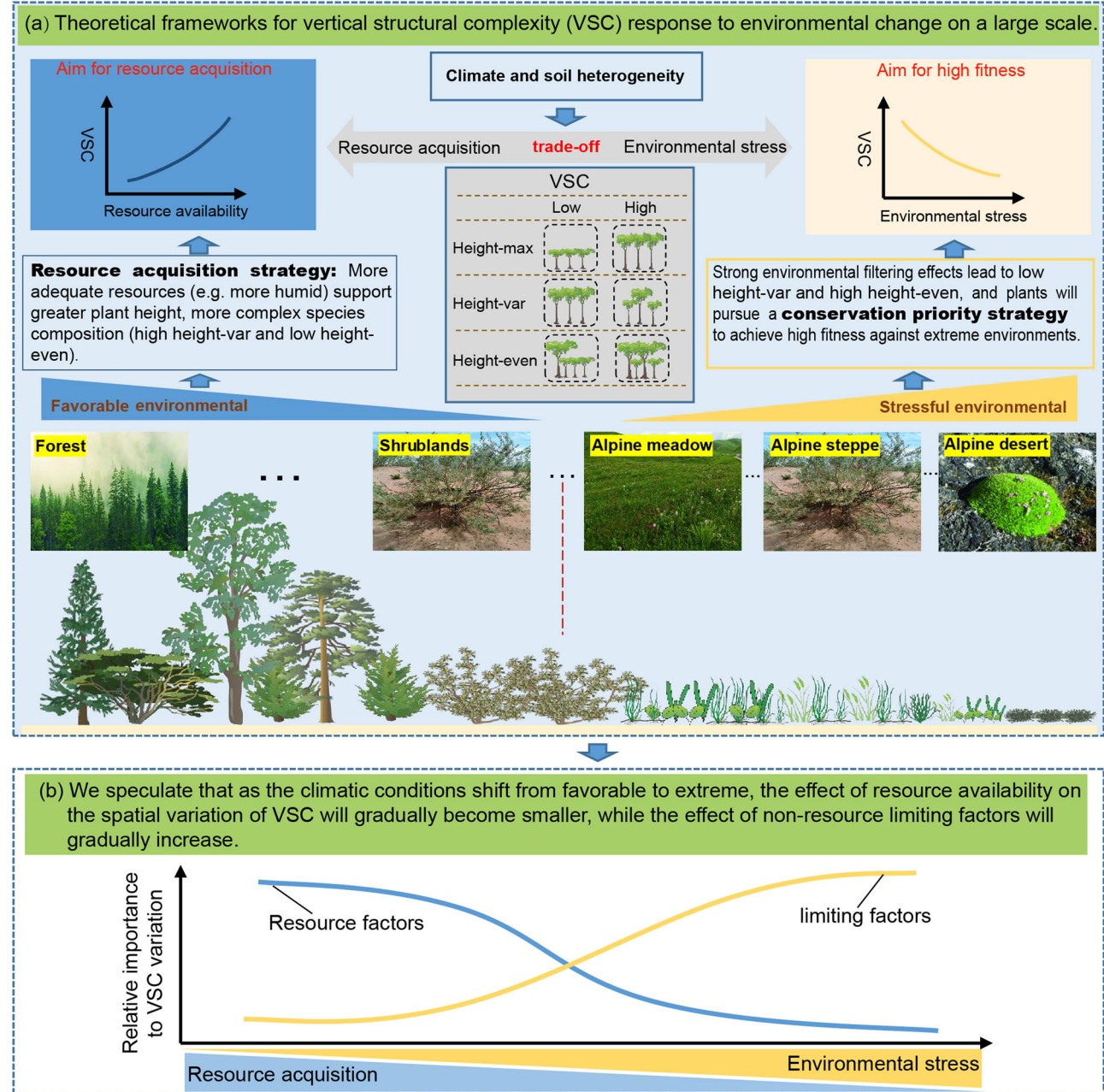

**Fig. 1 | . The theoretical framework (a) and scientific hypothesis (b) for the vertical structural complexity (VSC) response to environmental change on a large scale.** Under favorable environmental conditions, adequate resources supporting greater height and more complex community composition, as explained by the physiological tolerance hypothesis. In contrast, plants in regions of high environmental stress, strong filtering, and conservation priority will result in shorter and more homogeneous community structure. We, therefore, assumed that with the shift of climate from favorable to extreme, the effect of resource availability on VSC would become smaller, but the effect of non-resource limiting factors would gradually increase, as an adaptation mechanism. Height-max, maximum plant height within a plot; Height-var, coefficient of variation of plant height; Height-even, Shannon evenness of plant height.

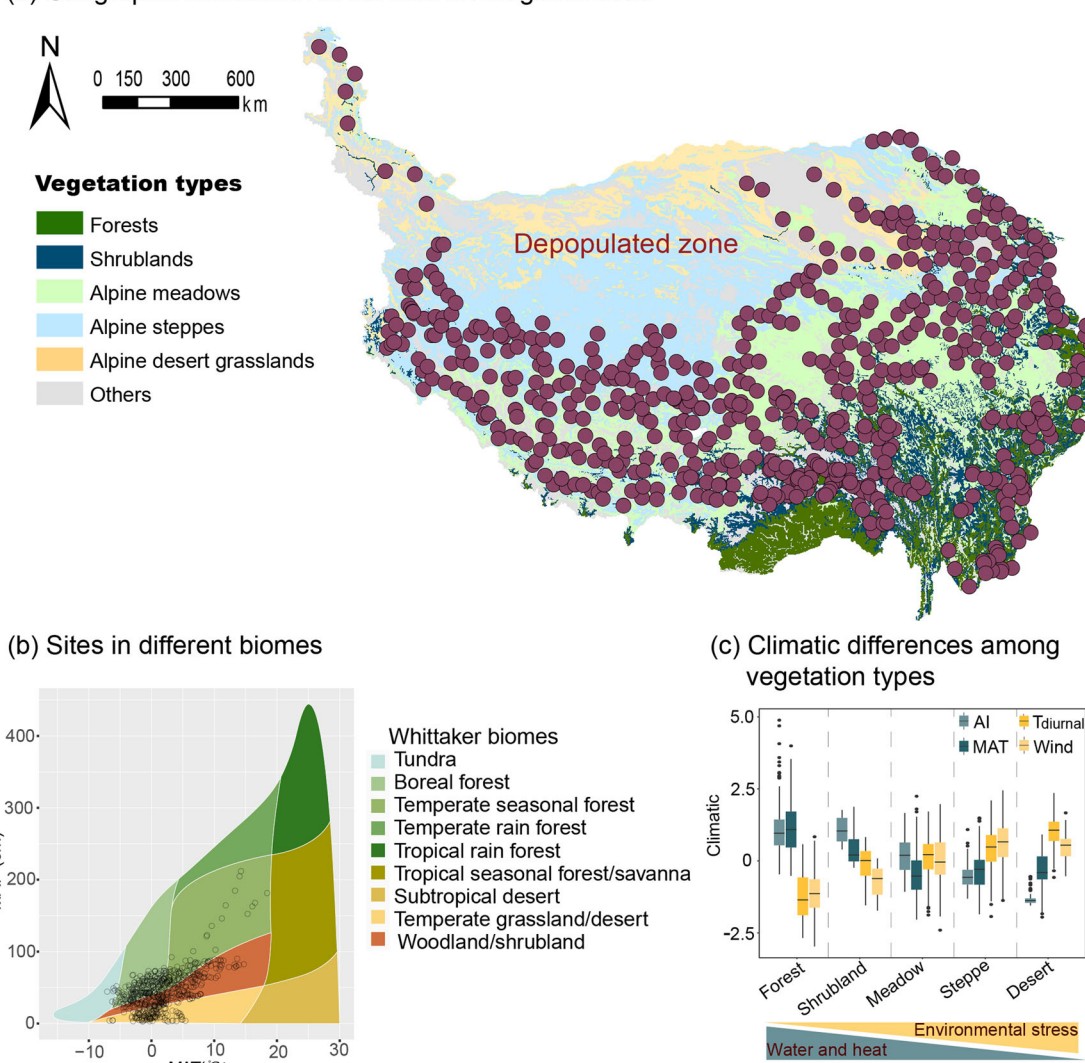

**(a) Geographic distribution of the field investigation sites**

**(b) Sites in different biomes**

**(c) Climatic differences among vegetation types**

**Fig. 2 | Geographical and spatial distribution of sampling sites and climatic differences among vegetation types of Tibetan Plateau (TP).** The background in figure (**a**) represents different vegetation types on the TP. The biome types were defined in Whittaker's classification (**b**). The climatic data in figure (**c**) was standardized to mean = 0 and standard deviation = 1 to aid their presentation in one graph. The black lines represent the median. The depopulated zone was not investigated due to a lack of access. MAT annual mean temperature, AI aridity index, T$_{diurnal}$ diurnal temperature range, Wind wind speed.

availability is low and environmental stresses are high)[14,16–18]. Therefore, we assumed that as climatic conditions shift from favorable to extreme, the impact of resource availability on spatial variation of VSC will decrease as environmental stress increases, while the effect size of non-resource limiting factors (that is, environmental factors that cause physiological restrictions on plant growth and reproduction without providing resources) such as extreme low temperatures, high ultraviolet radiation, and strong wind will gradually increase (Fig. 1). When resource availability is higher (e.g., favorable precipitation and heat conditions), the resource acquisition strategy of plants dominates VSC variation (emphasized by PTH). When environmental stresses are high, the environmental filtering effects of non-resource limiting factors will be strong, implying a simpler community composition under harsher environmental conditions[19–21]. In addition, plants that have adapted to harsh environments should pursue a conservation strategy (e.g., morphological changes in plants and facilitations) to avoid mortality[16,22]. For example, under conditions of extremely low temperatures and strong winds, plants will maintain a suitable temperature in their immediate environment by growing near other plants and growing closer to the ground, which is similar to the idea of "warming in a group"[23]. This dwarf plant community avoids freezing damage caused by low

temperatures and wind-shearing forces caused by strong winds[23,24]. Although these theoretical underpinnings provide the basis for our extrapolations, but few empirical studies have been able to span from the favorable to extreme climate gradient.

The Tibetan Plateau is an ideal region for investigating the spatial patterns and underlying mechanisms of VSC, because it has a wide range of biomes from subtropical forests to tundra, a consequence of the variation in environments resulting from the broad altitudinal range in the region (Fig. 2b). Considering this variation, it is also an important ecological reserve and has always been a hotspot for ecologists. We surveyed 2013 standard plots using a grid-sampling method, covering five major vegetation types in the region (Fig. 2a). Three parameters were selected to quantify VSC, viz. maximum plant height within a plot (Height-max), the coefficient of variation of plant height (Height-var), and the Shannon evenness of plant height (Height-even). These three parameters can fully describe the spatial niche occupancy of community individuals in the vertical dimension[2]. The main objectives of this study were to: (1) explore the changes in VSC of different vegetation types on the Tibetan Plateau; (2) identify the factors influencing VSC spatial variation; and (3) produce 1-km spatial resolution maps of VSC, which could provide important parameters for related

**Table 1 | Differences in vertical structural complexity among different vegetation types of the Tibetan Plateau**

| Vegetation type | Height-max /m | | Height-var | | Height-even | | $N^a$ |
|---|---|---|---|---|---|---|---|
| | Mean | SD | Mean | SD | Mean | SD | |
| Forests | 23.48 | 8.11 | 0.38 | 0.13 | 0.74 | 0.13 | 456 |
| Shrublands | 1.23 | 0.51 | 0.79 | 0.18 | 0.75 | 0.11 | 30 |
| Alpine meadows | 0.17 | 0.11 | 0.75 | 0.35 | 0.69 | 0.10 | 669 |
| Alpine steppes | 0.16 | 0.08 | 0.67 | 0.31 | 0.75 | 0.12 | 621 |
| Alpine deserts | 0.17 | 0.10 | 0.64 | 0.32 | 0.78 | 0.13 | 237 |

$^a N$ number of plots.

macroecological studies. Specifically, we tested our assumption that the underlying mechanisms that dominate VSC variation will gradually shift from resource acquisition strategies to conservation priority strategies as environmental conditions shift from favorable to extreme (Fig. 1), and that the classic PTH would be more applicable in "favorable conditions".

## Results

### Changes in VSC among different vegetation types on the Tibetan Plateau

The Height-max in the forests, shrublands, alpine meadows, alpine steppes and alpine desert grasslands were 23.48 (±8.11 SD), 1.23 (±0.51 SD), 0.17 (±0.11 SD), 0.16 (±0.08 SD) and 0.17 (±0.10 SD) m, respectively. The Height-var of the alpine meadows was larger than that of the alpine steppes, followed by the alpine desert grasslands, but the Height-even of the alpine desert grasslands was greater than that of the alpine steppes, followed by alpine meadows (Table 1).

### Spatial patterns of VSC on the Tibetan Plateau

The Height-max decreases with latitude gradient, and communities growing between 26–29° N were 93 times taller than those growing between 38–41° N (Fig. 3). The Height-max increased and then decreased with the altitude gradient and was the tallest at 1000–2000 m. Furthermore, the mean Height-var was smaller between 38–41° N and 5000–6000 m than elsewhere. No clear pattern of variation in Height-even along latitude and elevation was observed (Fig. 3). From the southeast to the northwest of the Tibetan Plateau, the Height-max decreases while the Height-even increases (Fig. 4a, c). The Height-var was lowest in the central regions of the Tibetan Plateau (Fig. 4b).

### Environmental factors influencing VSC

For Height-max and Height-var, aridity index was the most influential factor in forests and shrublands (Table 2) and increased with precipitation availability (Fig. S1). However, in alpine grassland ecosystems (including alpine meadows, alpine steppes, and alpine desert grasslands), the non-resource limiting factors (especially diurnal temperature range [$T_{diurnal}$], wind, and oxygen partial pressure [$PO_2$]) played a more important role in the spatial variation of Height-max and Height-var (Table 2). The stronger the environmental stress, the smaller the Height-max and Height-var (Fig. S1). For Height-even, Aridity index was the dominant factor in shrublands ($R^2 = 0.669$), and wind explained 11.0% of the variation in alpine meadows. However, in the analysis of forests, alpine steppes, and alpine desert grasslands, the explanatory power of a single variable for Height-even variation was low (less than 10%) (Table 2).

Furthermore, the spatial autocorrelation value of the model residuals was very close to zero, indicating that the effect of spatial autocorrelation on the multiple stepwise regression results was negligible (Fig. S2). Furthermore, the results of the partial correlation analysis showed that the zero-order correlation coefficient was not significantly different from the partial correlation coefficient, indicating that the grassland community sampling method had a negligible effect (Table S2).

### Potential mechanisms for the spatial variation in VSC

Summary based on multiple regression in Table 2 (original samples) and Table S3 (random samples), when climatic conditions gradually change from favorable to extreme (transition from forest to alpine grassland ecosystem), the effects of resource availability on the variation in VSC were gradually declined from subtropical forests to alpine desert grasslands, and the effects of non-resource limiting factors have become larger (Fig. 5).

### High-resolution maps of VSC on the Tibetan Plateau

Based on machine learning models, maps of Height-max, Height-var, and Height-even with 1-km resolution were first produced on the Tibetan Plateau (Fig. 4). The models explained 85%, 55%, and 33% of the spatial variations in Height-max, Height-var, and Height-even, respectively (Fig. S3). Annual temperature range ($T_{annual}$) was the most important predictor of LogHeight-max spatial variation, followed by oil organic carbon (SOC) and $T_{diurnal}$. Wind was the most important predictor of Height-var spatial variation, followed by UR and minimum temperature of the coldest month ($T_{coldest}$). Wind was the most important predictor of Height-even spatial variation, followed by mean annual precipitation (MAP) and ultraviolet radiation (UR). To test the model, 25% of the randomly selected plot data were used. The model predictions of Height-max, Height-var and, Height-even were accurate, because the observed and predicted values were closely and evenly distributed on both sides of the 1:1 line (Fig. 4).

## Discussion

Based on large-scale field survey data on the Tibetan Plateau, this study explains the determining mechanism of spatial variation in VSC, and produces 1-km resolution spatial atlas based on machine learning models. We found that Height-max decreases with latitude, which is consistent with the global study by Moles, et al. (2016)[25], and further deepens our knowledge that plants are smaller at high latitudes[26]. Plant height first increased and then decreased along the altitudinal gradient, which is inconsistent with the monotonically decreasing trend found by Mao, et al. (2016)[27] on the Tibetan Plateau, and is likely due to optimal precipitation and temperature conditions at mid-altitude regions[28]. We also found that Height-var was lowest in the regions of maximum altitude (5000–6000 m) and highest latitude (38–41°N). These findings confirmed our inference that plant communities possess shorter and more uniform vertical structures under extreme environmental conditions. We found that aridity index was the dominant factor in the spatial variation of Height-max and Height-var in forest and shrubland ecosystems, and wetter environments supported higher and more complex community structures (Table 2, Fig. S1). These findings support classic PTH[10,11,29]. These results highlight the importance of considering future changes in water availability in forest and shrubland ecosystems, with predictions that the forest and shrubland communities in the eastern part of the Tibetan Plateau will become taller and more complex under wetter and warmer conditions in the Tibetan Plateau[30].

As expected, resource availability had weak explanatory power in alpine ecosystems, such as alpine meadows, alpine steppes, and alpine desert grasslands, where the influence of non-resource limiting factors became stronger (Table 2, Fig. 5). For the alpine steppes and alpine desert grasslands on the Tibetan Plateau, the most influential factor on Height-max was $T_{diurnal}$ (Table 2). In the context of global warming, we predict that plants in the alpine steppes and alpine desert grasslands will become taller, as the $T_{diurnal}$ will become smaller due to the warmer conditions at night than during the day[31], thus reducing the constraints on plant growth. This is consistent with the prediction by Olson, et al.[32] that plant height in the Arctic tundra will increase under global warming. Furthermore, Wind (non-resource limiting factors) was an important predictor of Height-var and Height-even in alpine meadows, alpine steppes, and alpine deserts (Table 2). In the context of climate change, there is no consensus on how the wind speed of the Tibetan Plateau will change, and both increases[33] and decreases[34] have been predicted. Therefore, how the Height-var and Height-even of alpine grasslands will change in the future is uncertain.

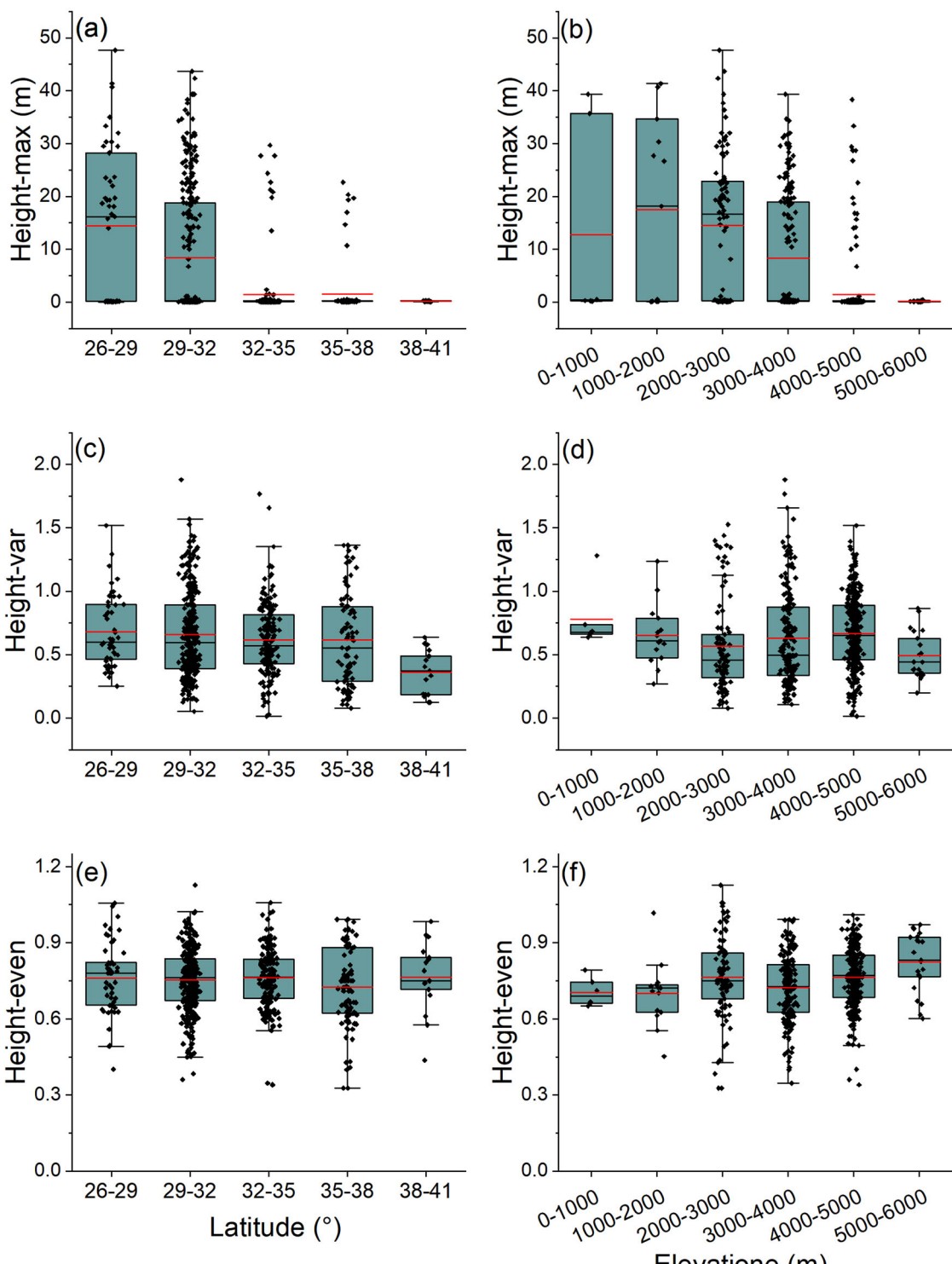

**Fig. 3 | Response of vertical structural complexity to latitude and elevation. a, c, e** Latitude; **b, d, f** elevation. The red lines represent the mean. Height-max, maximum plant height within a plot; Height-var, coefficient of variation of plant height; Height-even, Shannon evenness of plant height.

Considering a wider climatic gradient and different vegetation types, the spatial variation of VSC should be the result of the trade-off between acquiring resources and achieving greater fitness to avoid mortality (Figs. 5, 6), which is unlike the predictions based on PTH[10–12,29]. The potential mechanisms of non-resource limiting factors on VSC spatial variation could be explained as follows. First, the environmental filtering effect selects species that can survive in alpine habitats, which implies a simpler community composition under harsher environmental conditions[10,35]. Second, these plant communities are tightly packed and grow close to the ground, thereby reducing their exposure to extreme climates. This is a survival strategy being similar to "warming in a group"[23]. Third, under extreme environmental conditions, competition between plants is weakened, and the facilitation will be enhanced for resistance to adverse environmental conditions; compared with competition, facilitation can tolerate the overlap in spatial niches between individuals and lead to a more uniform community structure[18].

**Fig. 4 | Predicted spatial patterns of VSC on the Tibetan Plateau at a resolution of 1 km.** Panels (**A**), (**B**), and (**C**) are the spatial distribution of Height-max, Height-var, and Height-even, respectively; panels (**a**), (**b**), and (**c**) are the relationship between the observed and predicted values, using the Random Forest model. The black dotted line is the 1:1 line, and $R^2$ represents the prediction accuracy of the random forest models. Height-max, maximum plant height within a plot; Height-var, coefficient of variation of plant height; Height-even, Shannon evenness of plant height.

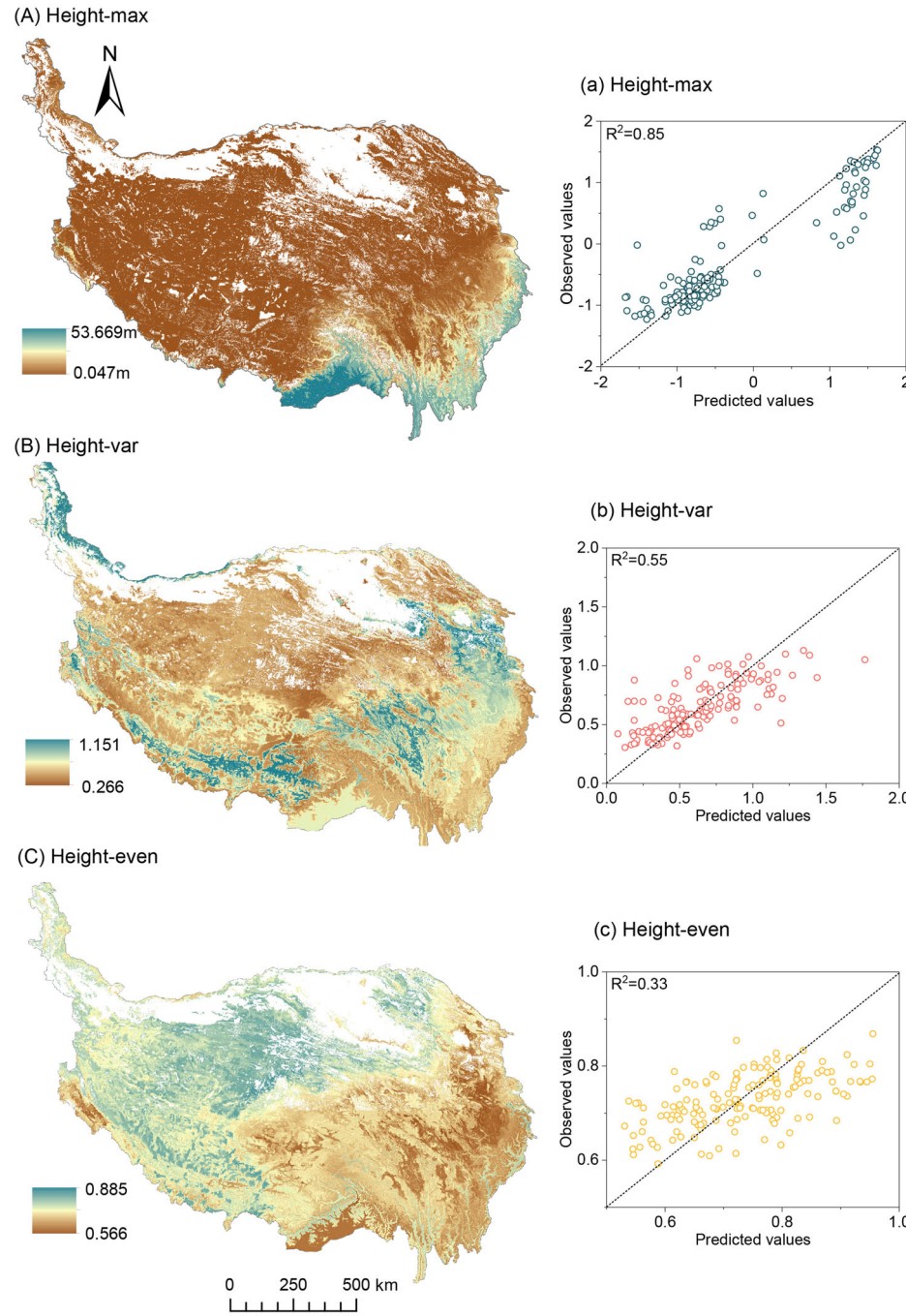

We produced 1-km resolution maps of the VSC on the Tibetan Plateau for the first time (Fig. 4). These maps can provide important parameters for macroecological studies from many perspectives, such as the interpretation of underlying mechanisms of the diversity-productivity relationship, more accurate prediction of productivity variation, and the modification of carbon cycle models[15]. VSC has extensive connections with ecosystem processes, so its atlas may play important value in other aspects, and its role needs to be further explored in the future. Specifically, the general connections between VSC and ecosystem processes are as follows: (i) higher vegetation heights will promote primary productivity, carbon sequestration, and landscape heat uptake and more complex vertical structures will facilitate community function by enhancing spatial complementarity effects[3,32]; (ii) higher and more complex vertical structures can promote the coexistence of species at other trophic levels by providing more ecological niches[25]; and (iii) complex vertical structures mean communities have

diverse physiological traits (such as shade tolerance, crown plasticity, etc.), which will affect the functional diversity of the community and thus the multifunctionality of the entire ecosystem[25] (Fig. 6).

Despite the significant advantages and high prediction ability of these maps (Fig. 4), it is a long method that requires improvement. There are few limitations of the present study. First, several areas were under-sampled in this study, such as the depopulated zone in the northern region of the Tibetan Plateau, which may have resulted in some uncertainty in the regional estimate. Second, the machine learning model has the advantage of representing heterogeneously distributed observations, because it contains multiple predictor variables to represent spatial variation in climatic and soil properties[36]. However, it is the first to estimate these parameters of the VSC at a regional scale and requires further development in the future. Furthermore, the prediction for Height-even is relatively weak ($R^2 = 0.33$) (Fig. S3). In addition, plant communities in different successional stages

**Table 2 | Multiple stepwise regression models of the relationship between vertical structural complexity and environmental variables for five vegetation types on the Tibetan Plateau**

### Height-max[a]

| Vegetation | Variable | P[c] | VIF[b] | R² |
|---|---|---|---|---|
| Forest | AI | <0.001 | 1.136 | 0.352 |
| | Wind | 0.003 | 2.546 | 0.051 |
| | MAT | 0.057 | 2.713 | 0.027 |
| | | | | 0.43 |
| Shrublands | AI | 0.001 | 2.874 | 0.603 |
| | Wind | 0.012 | 2.234 | 0.18 |
| | pH | 0.204 | 2.792 | 0.078 |
| | | | | 0.861 |
| Alpine meadow | PO₂ | <0.001 | 1.86 | 0.151 |
| | Wind | <0.001 | 4.133 | 0.110 |
| | T_diurnal | <0.001 | 1.296 | 0.074 |
| | MAT | 0.115 | 3.495 | 0.049 |
| | AI | 0.095 | 2.221 | 0.018 |
| | pH | 0.074 | 1.107 | 0.007 |
| | | | | 0.408 |
| Alpine steppe | T_diurnal | <0.001 | 1.163 | 0.39 |
| | AI | 0.124 | 1.229 | 0.058 |
| | Wind | <0.001 | 1.163 | 0.029 |
| | | | | 0.477 |
| Alpine desert grasslands | T_diurnal | <0.001 | 1.362 | 0.205 |
| | Wind | 0.008 | 1.168 | 0.111 |
| | AI | 0.135 | 1.288 | 0.015 |
| | | | | 0.33 |

### Height-var

| Vegetation | Variable | P | VIF | R² |
|---|---|---|---|---|
| Forest | AI | <0.001 | 1.347 | 0.398 |
| | MAT | <0.001 | 4.972 | 0.099 |
| | pH | 0.117 | 1.28 | 0.021 |
| | UR | 0.037 | 1.4 | 0.012 |
| | PO₂ | 0.002 | 5.343 | 0.059 |
| | | | | 0.589 |
| Shrublands | AI | 0.025 | 1.987 | 0.403 |
| | Wind | 0.083 | 1.987 | 0.135 |
| | | | | 0.538 |
| Alpine meadow | MAT | <0.001 | 3.445 | 0.350 |
| | Wind | <0.001 | 4.122 | 0.231 |
| | PO₂ | <0.001 | 1.858 | 0.065 |
| | AI | 0.06 | 1.81 | 0.018 |
| | | | | 0.664 |
| Alpine steppe | MAT | <0.001 | 2.536 | 0.233 |
| | Wind | <0.001 | 3.629 | 0.186 |
| | PO₂ | <0.001 | 2.074 | 0.044 |
| | AI | 0.008 | 1.269 | 0.023 |
| | SOC | 0.037 | 1.175 | 0.019 |
| | | | | 0.506 |
| Alpine desert grasslands | Wind | <0.001 | 1.893 | 0.174 |
| | UR | <0.001 | 1.534 | 0.151 |
| | T_diurnal | 0.007 | 1.562 | 0.091 |
| | MAT | 0.079 | 2.175 | 0.029 |
| | | | | 0.445 |

### Height-even

| Vegetation | Variable | P | VIF | R² |
|---|---|---|---|---|
| Forest | AI | <0.001 | 1.221 | 0.104 |
| | UR | <0.001 | 1.107 | 0.032 |
| | T_diurnal | 0.114 | 1.329 | 0.019 |
| | Wind | <0.001 | 1.316 | 0.074 |
| | | | | 0.229 |
| Shrublands | AI | 0.001 | 1.726 | 0.669 |
| | T_diurnal | 0.012 | 1.46 | 0.151 |
| | SOC | 0.235 | 1.224 | 0.034 |
| | | | | 0.854 |
| Alpine meadow | Wind | <0.001 | 1.06 | 0.11 |
| | SOC | 0.098 | 13.03 | 0.007 |
| | TN | 0.102 | 13.164 | 0.006 |
| | pH | 0.161 | 1.114 | 0.004 |
| | | | | 0.127 |
| Alpine steppe | TN | <0.001 | 1.171 | 0.083 |
| | MAT | <0.001 | 1.005 | 0.071 |
| | UR | 0.032 | 1.166 | 0.04 |
| | | | | 0.194 |
| Alpine desert grasslands | Wind | 0.058 | - | - |
| | | | | 0.046 |

[a]Height-max, maximum plant height; MAT annual mean temperature, AI aridity index, Wind wind speed, PO₂ atmospheric oxygen partial pressure, UR ultraviolet radiation, T_diurnal diurnal temperature range, SOC soil organic carbon, TN soil total nitrogen content; Height-var, coefficient of variation of plant height; Height-even, Shannon evenness of plant height.
[b]VIF variance inflation factor.
[c]All models were significant at p < 0.001. R² for each model and its independent variables are provided separately.

**Fig. 5 | The relative importance (% of model R²) of resource factors and non-resource limiting factors for each vegetation type model.** The analysis of Height-even is not provided here, because the model we constructed is less explanatory (Table 2); see Table 2 and Table S2 for more detailed information. Height-max, maximum plant height within a plot; Height-var, coefficient of variation of plant height; Height-even, Shannon evenness of plant height.

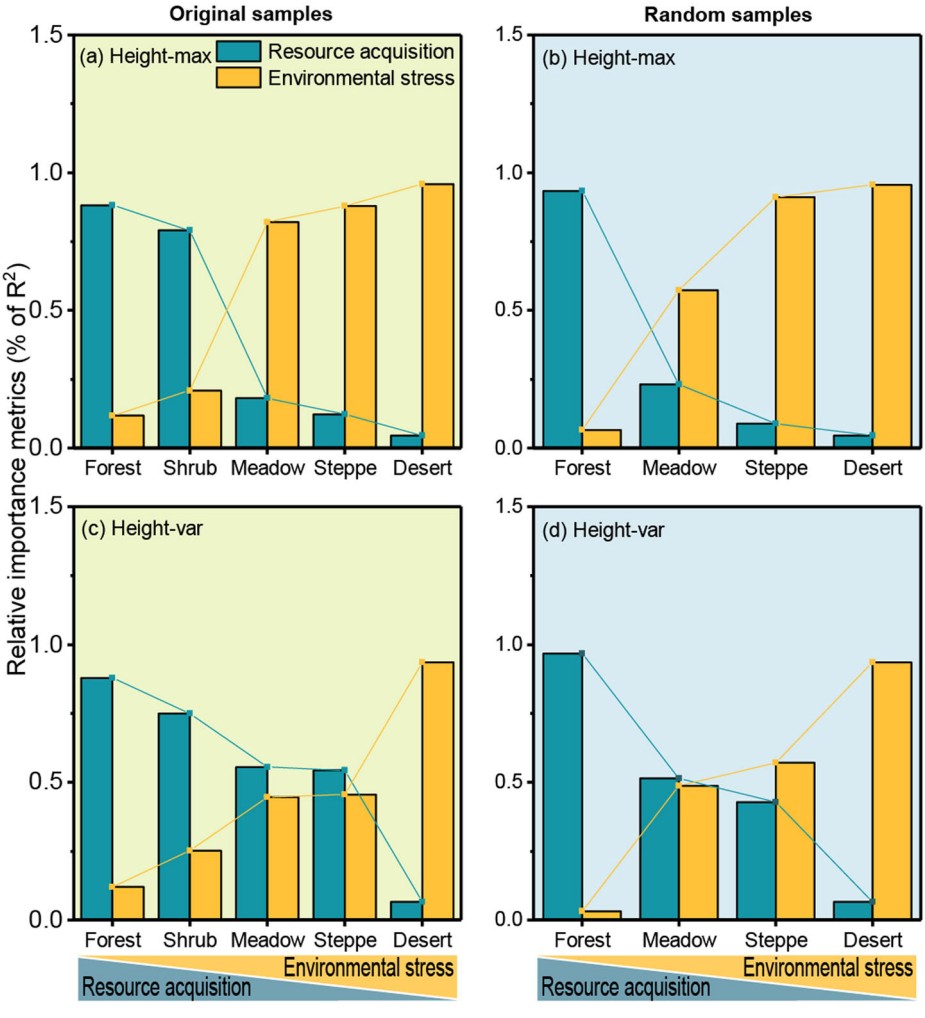

have different resource utilization and acquisition strategies, however, this study failed to consider this during the model construction process due to lack of data[9]. More importantly, with technological innovation, we predict that high-resolution remote sensing and terrestrial LiDAR will enable related research to be carried out on a larger scale and provide more useful information for responding to global climate change (Fig. 6).

## Methods
### Study area
The Tibetan Plateau is the largest ($2.5 \times 10^6$ km²) and highest plateau in the world[37,38]. Its unique topography creates a distinctively variable climatic gradient that provides excellent conditions for validating our hypotheses. The elevation of the Tibetan Plateau ranged from 80 to 8535 m, mean annual temperature (MAT) ranged from −23 to 24 °C, and mean annual precipitation (MAP) ranged from 18 to 3200 mm. From southeast to northwest, the precipitation and temperature conditions gradually declined, and the effect of non-resource limiting abiotic factors (e.g., extreme low temperature, diurnal temperature range [$T_{diurnal}$], UR, oxygen partial pressure [$PO_2$], and wind speed [Wind]) gradually increased (Fig. S4).

The Tibetan Plateau mainly includes five vegetation types that vary with climate: subtropical forests, shrublands, alpine meadows, alpine steppes, and alpine desert grasslands[39]. In previous studies, ecologists often divided temperature, precipitation, etc. into multiple environmental levels to explore the variation in biological responses along environmental gradients[40]. To make the division in the study ecologically meaningful, segmentation was determined at the boundaries between vegetation types as far as possible[40]. Exciting, the Tibetan Plateau has a unique advantage in this

regard, as the five vegetation types mentioned above correspond closely to the climatic gradients of the Tibetan Plateau. From subtropical forests to alpine desert grasslands, non-resource limiting factors such as $T_{diurnal}$ and Wind gradually strengthen, and the precipitation and temperature conditions gradually worsen (Fig. 2c).

### Field sampling
Field sampling was conducted during the high-growth period from mid-July to late August in 2018, 2019, and 2020. According to the latitude and longitude, we divided the entire Tibetan Plateau into 1000 grids of equal area (0.5° × 0.5°). For alpine grasslands (i.e., alpine meadows, alpine steppes, and alpine desert grasslands), we selected the dominant plant communities via visual observation in each grid and randomly set up three 1 m × 1 m plots for the field investigation. There were 1527 plots for grassland communities (Table 1). We determined the tallest plant in each plot by visual inspection, and the height of the tallest plant in each plot was measured with a steel ruler. For each species within the plots, we recorded their coverage and then randomly selected three plants of each species to measure their height with a steel ruler (all measurements for less than three individuals); in total, 19,320 plants were measured.

For forest and shrubland ecosystems, three 20 m × 20 m plots were randomly established in each grid, for a total of 486 plots (Forests: 456; shrublands: 30; Table 1). For forests, we recorded all vascular plants in these plots, including trees, shrubs, and herbaceous species. Diameter at breast height (DBH) was measured for trees with a DBH of ˃3 cm. A telescopic stick was used to measure the tree height, and 17,443 trees were measured. The maximum measurement scale of the telescopic stick is 20 m, and the

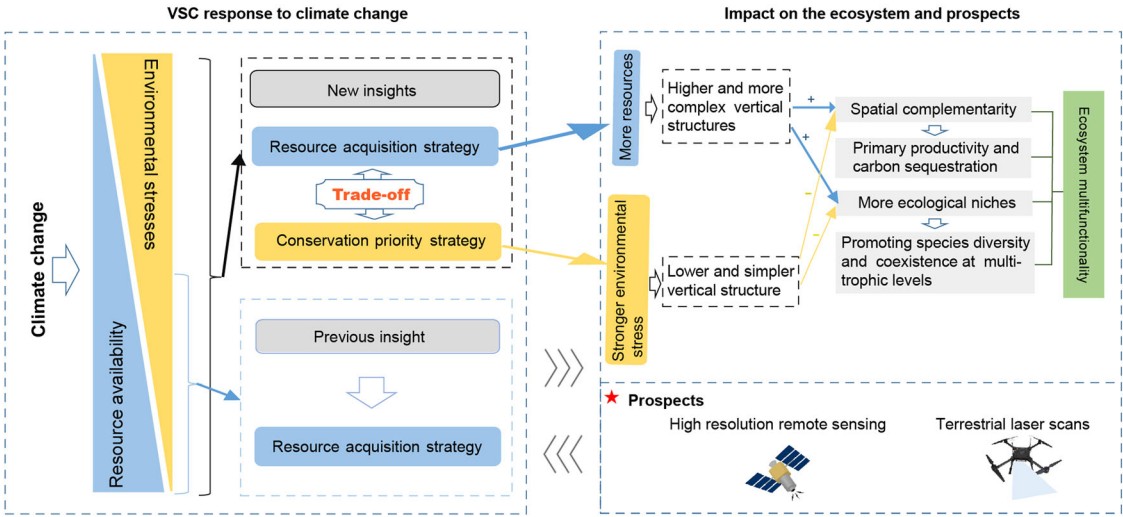

**Fig. 6 | Conceptual diagram of the response of community vertical structural complexity (VSC) to climate change.** VSC may also affect overall ecosystem function and stability, and future coping strategies.

telescopic stick is marked with measurements accurate to centimeters. For those trees with a height greater than 20 m, we complete the measurement based on climbing the trunk. Only trees with a DBH > 3 cm were measured, as this is a common standard for recording tree layer-related data in forest community surveys[41]. For shrublands, we recorded their basal diameter and height, and 339 plants were measured. Across all vegetation types, a total of 2013 field plots were investigated, and the height of 37,102 plants was used for the analysis. The area within the white curve in Fig. 2a is the depopulated zone, which was not investigated due to challenges accessing these sites.

## Calculations of VSC parameters

Three parameters were used to quantify VSC for each plot, where Height-max is the maximum plant height within a plot, and Height-var is the coefficient of variation of the plant height, which is calculated using Eq. 1. The Shannon evenness of plant height (Height-even) was calculated using Eq. 2[1]. The smaller the Height-var and the larger the Height-even, the more uniform the height distribution of the species in the plant community.

$$Height - \mathrm{var} = \frac{SD_H}{Mean_H} \qquad (1)$$

$$Height - even = \left[ -\sum_{k=1}^{Nh} Pk \times \ln(Pk) \right] / \ln(Nh) \qquad (2)$$

where $SD_H$ is the standard deviation of plant heights, $Mean_H$ is the mean plant height, $Nh$ number of height classes, and we used class widths of 100 cm, 10 cm, and 1 cm for forests, shrublands, and grasslands, respectively. For forests and shrublands, $Pk$ refers to the proportion of the basal area for the kth height class. For grasslands, we divided $Nh$ by the mean height of each species, and $Pk$ refers to the mean of the relative height and relative coverage of the kth height class.

## Selected explanatory variables

Based on the research objectives and hypotheses, 13 explanatory variables in total, representing resource- and non-resource limiting variables, were selected. Resource variables are characterized as precipitation conditions, heat conditions, and variables related to soil fertility, as they are often used to reflect an area's capacity and resources for plant growth, reproduction, and survival[10,35,42]. We chose MAP and an aridity index (AI) to represent precipitation conditions, while MAT represents heat conditions. We measured pH, soil organic carbon (SOC), and total nitrogen (TN) levels. SOC and TN are important indicators of soil fertility. Soil pH was considered because

highly acidic soil affects community species diversity and regulates soil nutrient supply and uptake by plants[42]. Growing-season climate variables were not considered, because it is difficult to define a consistent growing season over such a large spatial scale. Photosynthetically active radiation was also not considered, although competition for light resources within local communities is an important source of VSC. It has not been considered a resource factor in large-scale studies, because plants only use a very small part of the photosynthetically active radiation reaching the Earth's surface, generally considered to be less than 1%[20,43].

Non-resource limiting variables limit plant growth and reproduction[19]. As the third pole of the world, alpine ecosystems in the Tibetan Plateau's central and northwest regions are usually adapted to stressful environmental conditions. Based on prior knowledge[14,21,23,44], we chose the following seven variables: minimum temperature of the coldest month ($T_{coldest}$), diurnal temperature range ($T_{diurnal}$), annual temperature range ($T_{annual}$), atmospheric oxygen partial pressure ($PO_2$), atmospheric carbon dioxide partial pressure ($PCO_2$), ultraviolet radiation (UR), and Wind.

One topsoil sample (0–10 cm) was collected using an auger in each 1 m² grassland plot. Further, 15–30 topsoil samples were randomly sampled from each 400 m² forest and shrubland plot and then mixed into a composite sample[45]. The soil samples were air-dried and sieved (<2 mm) to homogenize them. Soil pH was measured using a pH electrode (Leici), SOC content was measured using the $H_2SO_4$-$K_2Cr_2O_7$ oxidation method, and TN was measured using Kjeldahl nitrogen determination.

MAT, MAP, $T_{coldest}$, $T_{diurnal}$, $T_{annual}$, and Wind data were downloaded from the WorldClim database (www.worldclim.org) at a spatial resolution of 30 arc-sec. The mean annual potential evapotranspiration (PET), with a spatial resolution of 30 arc-sec, was obtained from the Consortium of International Agricultural Research Centers (http://www.cgiar-csi.org/). AI is the ratio of MAP to PET[46]. Annual average UR was obtained from the Science Data Bank (https://doi.org/10.11922/sciencedb.332). $PO_2$ and $PCO_2$ were calculated according to the method of Kouwenberg, et al.[47].

## Statistics and reproducibility

A machine learning model (random forest model) was used to upscale the site-level VSC to the entire Tibetan Plateau under contemporary climate scenarios. The predictors included all 13 explanatory variables considered in this study. We used 75% of the total data as the training data and the remaining 25% as the validation data. The increase in node purity of the splitting variables was used to estimate the relative importance of the explanatory variables. Based on the spatially gridded data of the predictors (3,617,619 values), the spatial distributions of the VSC for each grid were mapped at a resolution of $1 \times 1$ km across the Tibetan Plateau.

We interpolated the pH, SOC, TN, $PO_2$, and $PCO_2$ at a spatial resolution of 1-km using "ordinary Kriging interpolation," which has the advantage of accurate interpolations at sampling locations[48]. Glaciers and lakes were removed from the maps. The Height-max was log-transformed to avoid the influence of data dispersion. The analyses were conducted using MATLAB 2018b (MathWorks, Natick, MA, USA) and the ESRI ArcGIS software (Version 10.2; Redlands, CA, USA).

Multiple stepwise regression (MSR) was used to determine the minimal adequate model, and the "calc.relimp" function in the R package *relaimpo* was used to estimate the relative importance ($R^2$ and relative importance metrics[%]) of the two types of variables in the model[49]. We performed an MSR for the five vegetation types. From subtropical forests to alpine desert grasslands, if the relative importance of resource variables gradually decreases and non-resource limiting variables increase, our assumption is supported (that is, the underlying mechanisms that dominate VSC variation will gradually shift from resource acquisition strategies to conservation priority strategies). Considering the differences in sample size may have an impact on the results. We randomly sampled the four vegetation types using the bootstrapping method and then repeated the above analysis. The shrublands were not considered in this process because it had only 30 plots. The sample size randomly selected was based on the sample size of the alpine desert grasslands (237 plots), because it has the smallest sample size of the remaining four vegetation types (Table 1).

When conducting MSR analysis, we screened explanatory variables based on the following principles. We further screened the 13 explanatory variables based on the Pearson correlation coefficient, because multicollinearity may affect the determination of the relative importance of factors. Five resource variables (MAT, AI, TN, SOC, and pH) and four non-resource limiting factors ($T_{diurnal}$, UR, wind, and $PO_2$) were screened. MAP was not considered, because its correlation with MAT (Pearson r = 0.62) was higher than that with AI and MAT (Pearson r = 0.45), and AI was strongly correlated with MAP (Pearson r = 0.95) (Fig. S1). We only retained $T_{diurnal}$ for low temperature-related variables, because it had the lowest correlation with MAT (Pearson r = −0.46) among all low temperature-related variables, and a strong correlation with $T_{annual}$ and $T_{coldest}$ (Pearson r = −0.71 and −0.60). $PCO_2$ was excluded because it was highly collinear with $PO_2$ (Pearson's r = 1). We further employed variance inflation factors (VIF) to verify whether the variables in the optimal model obtained by the MSR are independent[19]. We also calculated Moran's I values for both the observed VSC data and the residuals of the MSR models to examine how the spatial autocorrelation in the VSC was explained by the predictor variables[50]. MSR analysis was performed using the stepAIC function of the MASS package (http://www.biostathandbook.com/multipleregression.html).

Furthermore, we explored the impact of sampling methods on the results of grassland communities. Instead of measuring the height of each herb in the grasslands, we randomly selected three plants of each species. Therefore, the number and evenness of species within the communities may affect the analysis of Height-var and Height-even. To quantify the impact of this sampling, we employed partial correlation analysis. It was judged by whether the correlations (zero-order correlation coefficient) of Height-var, Height-even, and their dominant factors (based on MSR) changed after controlling for the effect of the Shannon-Wiener index (partial correlation coefficient). The important values (IV) of the Shannon Wiener index were calculated using the following formula[51]:

$$IV = (relative\ height + relative\ coverage)/2 \qquad (3)$$

Graphs were plotted using Origin software (version 8.5; Northampton, MA, USA). Statistical analysis was performed using R[52]. The significance level was set at $p < 0.05$.

## Reporting summary

Further information on research design is available in the Nature Portfolio Reporting Summary linked to this article.

## Data availability

All data for the figures and tables in the article are provided in the Supplementary Data.

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

## Acknowledgements

This work was supported by the Second Tibetan Plateau Scientific Expedition and Research Program (STEP, 2019QZKK060602), the National Natural Science Foundation of China [42141004, 42071303, 31988102], and China Science and Technology Cloud.

## Author contributions

Nianpeng He designed the research. Changjin Cheng and Jiahui Zhang conducted the research (collected the datasets and analyzed the data). Changjin Cheng wrote the manuscript. Mingxu Li, Li Xu, and Congcong Liu commented on the details of the manuscript drafts.

## Competing interests

The authors declare no competing interests.
