## [Peer review file · Communications Biology]

Reviewers' comments:

Reviewer #1 (Remarks to the Author):

Peer reviewing – Biology Communications COMMSBIO-23-0888-T

Manuscript title: " Vertical structural complexity of plant communities represents the combined effects of resource acquisition and environmental stress on the Tibetan Plateau"

The reviewed manuscript by Chen et al. investigates the shifting of climate gradient and resource availability effects on height-derived metrics (as a proxy for vertical structural complexity) of different ecosystems in the Tibetan Plateau (TP). More specifically, based on the physiological tolerance hypothesis, the authors expected to find plant height responses to environmental change conditions at a large scale (within five different vegetation types such as alpine meadows, alpine steppes, alpine desert grasslands, forests, and shrublands). In general, investigating what drives the VSC variation within ecosystems needs to be better understood (please see Coverdale and Davies 2023), particularly on a large scale. Hence, I found it exciting. Further, they used the different ecosystems model predictions to create an up-scaling map for the whole TP, displaying how vertical structural complexity changes under contemporary climate scenarios. Using height-derived metrics as a proxy for VSC is a quite typical approach to assessing vegetation response to environmental conditions and, therefore, not novel. However, I appreciated the sampling efforts in obtaining ground data on such a large scale. Note that this is relevant to deepen our understanding of the role of VSC on ecosystem functioning and to improve the predictability of ecosystem responses to future climate change scenarios (here with current climate scenarios, though).

General concerns:

The study needs further clarification, particularly on extending conceptual statements related to vertical structural complexity and its importance to the functioning of ecosystems. There are at least two decades of studies on VSC for different ecosystems (Ishii et al 2004; Gough et al 2019; Ehbrecht et al. 2021; McElhinny et al. 2005 and more recently Atkins, Bohrer, et al. 2018 and Coverdale et al 2023), and therefore a deepen revision and discussion about this critical ecological trait is necessary to be included here. The manuscript addresses two ecological concepts to explain how VSC can change over different ecosystems—first, the mechanisms of how biodiversity influences ecosystem functions by enhancing resource partitioning via high VSC. However, I found it quite vague as biodiversity effects are not addressed in the study or further discussed in the results. Resource acquisition strategies (via competition for resources) are the second most appropriate for this study. Therefore, I recommend re-structure the study's conceptualization accordantly to the study goals. Moreover, a clear explanation of why the specific height-derived metrics were selected as VSC proxies are necessary, as different VSC metrics might well respond to precipitation, temperature, and soil carbon changes.

I need help understanding the meaning of the terms used to define the gradient of climate shift from favorable to extreme. Regarding favorable, the authors used "resource availability," For extreme, the term was "non-resource limiting factors". The term referred to as the extreme needs to fully address its meaning, as non and limiting together many indicate more resource availability. I would remove the non and only use resource-limiting factors for extreme conditions.

However, I would be careful about statements related to the quantification of VSC as 20 years of studies have shown different possibilities from ground measurements via inventories or proximal sensing (TLS, drones, etc) and remote sensing (GEDI), even in large-scale studies.

The results section is short and needs substantial improvement. For example, the hypothesis was

presented as results (line 133-135 – verb tense) and, therefore, did not reasonable displaying of the outcomes, either explaining the potential mechanisms such as the shape and direction of changing the resource limiting factors. Moreover, the comparison between maximum height within latitudes over the globe (other studies – considering other ecosystems and vegetations type) and the TP, I could not follow as the reason why max height was selected was not fully presented. The discussion section follows the same pattern. The first paragraph brings a study using Altimeter data followed by comparisons, so it would be better to state the main findings accordingly to the expectations. Furthermore, a deeper discussion of why the aridity index was so important and what it means regarding different ecosystems within the TP. Finally, the discussion needs to address a compressive argumentation on the potential mechanism driving the variation on VSC on the TP. Some issues with the writing need much improvement, and please see the minor concerns.

Minor concerns

Line 31: the definition of VSC is not clearly stated. I would suggest looking at the review paper by Atkins, Fahey, et al., 2018 and other papers Bergen et al., 2009; McElhinny et al., 2005; and, Müllerová et al., 2021.

Line 34: Barry et al. 2019 proposes that complementarity and not only complementarity effects (as it is only a quantification ofoveryielding in mixture relative to monocultures) can happen due to different causes and one of them how species differ in the use of resources. This sentence is not clear stating what Barry et al. 2019 proposes, so I would re-write it in way that likely species that use resources differently may display different functional traits expressed in the VSC.

Line 37: is highlight the importance of light harvesting and light-use efficiency via VSC, but any radiation measurements were considered in this study, I would give another example.

Line 39-40: I would be careful with these statements (even using your own review) as cited before, the VSC have been deeply used even in large-scale remote sensing studies.

Line 43-45: Again ... TLS, GEDI are open-access data available and not coarse-grained atlases.

Line 49: is there any citation to support this statement?

Line 59: This sentence reads not clear: maybe a suggestion “PTH help to some extent the understanding of VSC spatial variation.”

Line 65: hard time to understand these terms: resource availability and non-resource limiting factors...

Line 65-66: will gradually become smaller with what? The sentence for me is not complete.

Line 67: complete the sentence. “will gradually increase VSC”

Line 87-89: why were they selected? not introduced before. Why can variation in height display VSC? Just follow the reasoning such as different patterns of tree height, can provide insights into habitat adaptation, coexistence, evolution, etc.

Line 115: use of abbreviation before the term has been introduced AI was not introduced before in this sequence of the manuscript that I’ve got it.

Line 131-137: This section is quite short and confusing as it states the hypothesis as the results. I guess the verb tense is not correctly used, but confirming assumptions is poorly explained.

Line 135: Fig 5 and then later comes Fig. 4 (line 140). So, the presentation and discussion of figures are not in sequence.

Line150: I would start the discussion with a wrap-up of the findings based on what were predicted.

Line 152: which broaföeaf evergreen forest? Can it also be a tropical forest? then 12 meters is too short. Please clearly state the area. a quick research of CHM of broadleaf and evergreen forest mean height - max can vary between 20 - 60 m in temperate zones...

<https://doi.org/10.3390/rs15040975>

<https://doi.org/10.3390/rs15061522>

Line 155: I was impressed with 23 meters maximum height measured with a telescopic stick. How

could it be done? The methods section did not explain exactly who the stick was used to measure the top canopy of the trees in the forests.

Line 155-156: there was no introduction to why max height was so important to measure.

Global average of broadleaf forest in Temperate to arctic zones... please be specific –

Line 157: be specific to your area; think about tall trees in the tropics, only precipitation is not the most important factor; light and temperature influence a lot.

Line 160: could compare with other alpine grasslands... European Alps?

Line 161- 162: why is it important to compare alpine grasses with arctic tundra dominated by lichens???

Line 163: the missing year in the citation

Line 164: 36% lower in maximum height?

Line 165- 166: global middle latitudes. So, I really do not understand this sentence.

What is the relation of the high latitudes' maximum height of TP with the lowest global regions?

Line 183: I can not understand this sentence. PTH is not the hypothesis? I think here the predictors' name is missing.

Line 186: how did Tdiurnal affect height max? state the directional relationship that the reader can try to follow your argumentation. Where can I see this? Fig?

Line 191: Throughout the text: why the variables have first capital letters in the text?

Line 192: why discuss climate change now? Why does not bring it to the point of the hypothesis?

Line 196: replace more for different.

Line 198: Why defense here? It was never introduced before.

Line 206: why the community will be more uniform if you have facilitation? It could well be a facilitation via complementarity and then they will be able to diversity in structure.

Line 213: I would think the opposite: biological diversity promotes the complex vertical structure of plant communities.

Line 244: wording Exciting

Line 246- 247: this is the first time limiting factors are defined as they seem to be.

Line 250-251: no temporal change results?

Line 255: How was max height measured with a stick?

Line 257: three plant individuals per species?

Line 256: Missing methods on how height was measured in grasslands.

Line 259-260: This sentence is confusing, please consider to rewriting it. For forests, there were 456 plots and shrubs 30 plots. How was the size of all plots?

The three plots 20 x 20 means what? for forests or for shrubs and why only 3 randomly?

Line 270: Is the height-max averaged per plot?

Line 288: what is MAP? Not defined before? as you did to AI.

Line 294: Light is super important. But not considered.

Line 296-297: Although the study considers a large area, they are still comparing different ecosystems and measuring at the local level. This argumentation is weak. Second, the argument here can be discussed (different kinds of literature cite other results), but the point is PAR is fundamental for VSC as it is directly related to the plant structure. Leaf angles, arrangement, and distribution across height strata of a plant. So, it is important for resource acquisition such as light, water and biomass production. This is just not fit for this study.

Line 299: This is citing Callaway, which found that in a short experiment, plant species displayed a strong shift from competitive interactions in a relatively good EC to facilitation processes in more stressful EC. So, this sentence is not fully correct with the cited paper.

Line 302: Different types of abbreviations. Please be consistent. Please write within the brackets for minimum temperature (MAT)

Line 307: sample randomly within an area of 400 m². How exactly was the area chosen?

Line 318: missing year of the citation

Line 338: First, remind the reader of your assumptions again.

Line 339: typo

Line 344: it is not after? It is just confusing further. I would move this paragraph before the MSR.

Reviewer #2 (Remarks to the Author):

Here, in this interesting study, the authors aim to explore how the plant community vertical structural complexity (VSC) vary and how the driving factors are and also establish a VSC map in Tibetan Plateau, China. It is surprising the author to explore these topics by field sampling of so large number of plots, a very difficult work as far as I know. The topic of this study is very clear and important, and the results are convincing based on such great dataset, and the writing is nice. I would not repeat the main results of this paper and give a few of comments after I completely reviewing this paper.

L18 and throughout this paper: It is better to use the full name of the Tibetan Plateau rather than the abbreviation of TP.

L53: I still wonder why the warmer and more humid sites have more plant species with different strategies. This is a generally phenomenon, and there are many mechanisms for why it occurs and I'd like to see your idea on this topic. As you also mentioned, the harsh sites usually accommodate few species, and it is easy to understand this; while how it will be more diverse of the plant community in the agreeable conditions. This is also a important question as when we looking into plant roots and the authors could refer to a paper by Laughlin et al., 2021 NEE.

L56-57" It is true that the big plants can reduce the small plants in a community, for example, and the big plants could be more. Then why such big plants dominated communities can lead to greater VSC? The authors could explain it more clearly.

L60-62: Here I understand the tradeoff refers to different plant communities in different environmental conditions rather than the same environment in different conditions, e.g., the favorable or the extreme conditions. Therefore, I suggest to deleting the "specific" in L60.

L66: It is cool using the non-resource limiting factors and resource limiting factors, and I like it.

L159: add a reference.

L205 "facilitation of common resistance" is confusing, please rephrase.

L215-216 Here the authors refer to the functional diversity and multifunctionality. Could you please explain these terms clearly and why the VSC can enhance these diversity as from its current form I do not understand why variation of the VSC result in the functional diversity change.

L217 Yeah, these maps are very important, and here I'd like to know whether the authors considering the temporal variation of the plant height, variance and evenness into the model as it is possible the communities may not all be in the climax stage of the community succession. This is just a suggestion and I hope the authors could take this point into their discussion.

Point-to-point responses to the comments

(COMMSBIO-23-0888-T: Vertical structural complexity of plant communities represents the combined effects of resource acquisition and environmental stress on the Tibetan Plateau)

Dear Editor and Reviewers,

Thank you very much for your encouragement and comments which greatly improved the paper. We have revised the manuscript thoroughly. Furthermore, we have invited the native English speaker in the company of editage (www.editage.cn), to polish our English usage.

Here, we provide four files to show how we revised the manuscript: 1) revised main file; 2) revised main file with tracked changes; 3) revised supplementary file; 4) point-to-point responses to reviewer's comments (attached below).

Thank you for considering this paper, which we have revised to be timely and exciting for Communications Biology.

Yours sincerely,

Nianpeng He

Institute of Geographic Sciences and Natural Resources Research, CAS

Datun Road A 11, Chaoyang District, Beijing 100101, P.R. China

E-mail: henp@igsnrr.ac.cn

Tel: 86-10-64889263

Fax: 86-10-64889432

Reviewer: 1

General concerns:

#1 Using height-derived metrics as a proxy for VSC is a quite typical approach to assessing vegetation response to environmental conditions and, therefore, not novel.

Response: We have made revisions. Quantifying VSC based on traditional quadrat survey data is indeed not a very novel way. This study mainly wants to emphasize that although the method of quantifying VSC based on traditional quadrat survey data is not novel, its spatial pattern and underlying mechanisms have been ignored on a large scale. Exploration in this area may help solve two very important problems at present, namely providing theoretical and data support for interpreting the underlying mechanism of diversity-productivity and accurately predicting spatial variation of productivity. Based on full consideration of your suggestions, we have made major changes to the introduction. Among them, we have revised the VSC quantification method part as follows: “Currently, there are two ways to quantify VSC, one is based on plant height data from traditional field surveys, and the other is the recently emerged technology such as airborne laser scanning⁹. Despite recent advances in airborne laser scanning-based VSC studies at the global scale, their usefulness is somewhat hampered by the limited number of sample sites⁹. At the same time, these relative studies are commonly limited into specific ecosystem, and the universal mechanisms of VSC response to environmental change and to a wide range of climatic gradients is unclear¹⁰. In contrast, millions of plant height data are stored in various databases (e.g., sPlot) and literatures. The idea of quantifying VSC based on traditional field survey data can easily be extended, thereby providing ideas for predicting spatial variation in productivity at the global scale.”(Line 49-58)

#2 The study needs further clarification, particularly on extending conceptual statements related to vertical structural complexity and its importance to the functioning of ecosystems. There are at least two decades of studies on VSC for different ecosystems (Ishii et al 2004; Gough et al 2019; Ehbrecht et al. 2021; McElhinny et al. 2005 and more recently Atkins, Bohrer, et al. 2018 and Coverdale et al 2023), and therefore a deepen revision and discussion about this critical ecological trait is necessary to be included here.

Response: We have revised the introduction. Specifically, in the first paragraph, we emphasized that the determination mechanism of VSC and its high-resolution atlas can provide key support for current research in two aspects. The first aspect is to interpret the potential mechanism of the diversity productivity relationship; the second aspect is a more accurate prediction of spatial variation in productivity. In the second paragraph, we describe the two methods currently used to quantify VSC, based on field survey data and lidar; in this paragraph we emphasize the necessity of using field survey data. We have revised as follows: “Vertical structural complexity (VSC) in plant heights, which is physical niche partitioning in above-ground space, is thought to be an important property for a specific plant community². VSC is closely linked to various ecological processes^{1,2}. For example, in research on the diversity-productivity relationships, the classical complementary effect³ proposes that the spatial complementarity due to VSC is a key determinant of overyielding in species-rich communities^{4,5,6}, as more complex vertical structures mean stronger spatial niche partitioning,

thereby reducing competition^{7,8} and leading to the unique occupancy of niche axes such as light^{1,4}. However, few studies on the diversity-productivity relationships provide direct evidence that VSC links diversity to productivity, mainly because relevant studies are rarely able to quantify VSC (globally, only 10 of 136 studies were quantified, and all were focused on local scales; Table S1). In addition, a recent study showed that VSC has a stronger ability to explain spatial variation in productivity than species diversity at regional scales⁹. Therefore, it is currently necessary to explore the underlying mechanisms of VSC spatial variation at the regional scale and to produce high-resolution VSC atlases, which can provide support for interpreting the underlying mechanisms of the diversity-productivity relationship and more accurately predicting the spatial variation of productivity, thereby improving the predictability of ecosystems to climate change.

Currently, there are two ways to quantify VSC, one is based on plant height data from traditional field surveys, and the other is the recently emerged technology such as airborne laser scanning⁹. Despite recent advances in airborne laser scanning-based VSC studies at the global scale, their usefulness is somewhat hampered by the limited number of sample sites⁹. At the same time, these relative studies are commonly limited into specific ecosystem, and the universal mechanisms of VSC response to environmental change and to a wide range of climatic gradients is unclear¹⁰. In contrast, millions of plant height data are stored in various databases (e.g., sPlot) and literatures. The idea of quantifying VSC based on traditional field survey data can easily be extended, thereby providing ideas for predicting spatial variation in productivity at the global scale.” (Line 32-58)

#3 I need help understanding the meaning of the terms used to define the gradient of climate shift from favorable to extreme. Regarding favorable, the authors used "resource availability," For extreme, the term was "non-resource limiting factors". The term referred to as the extreme needs to fully address its meaning, as non and limiting together many indicate more resource availability. I would remove the non and only use resource-limiting factors for extreme conditions.

Response: Because two reviewers had opposing opinions on using the term "non-resource limiting factors", I have defined "non-resource limiting factors" in this revision where it first appears: “non-resource limiting factors (that is, environmental factors that cause physiological restrictions on plant growth and reproduction without providing resources)” (Line 78-80)

#4 However, I would be careful about statements related to the quantification of VSC as 20 years of studies have shown different possibilities from ground measurements via inventories or proximal sensing (TLS, drones, etc) and remote sensing (GEDI), even in large-scale studies.

Response: In the revised version, we compare the advantages and disadvantages of the two methods and the necessity of using quadrat survey data for research. The main revisions are as follows: “Currently, there are two ways to quantify VSC, one is based on plant height data from traditional field surveys, and the other is the recently emerged technology such as airborne laser scanning⁹. Despite recent advances in airborne laser scanning-based VSC studies at the global scale, their usefulness is somewhat hampered by the limited number of sample sites⁹. At the same time, these relative studies are commonly limited into specific ecosystem, and the

universal mechanisms of VSC response to environmental change and to a wide range of climatic gradients is unclear¹⁰. In contrast, millions of plant height data are stored in various databases (e.g., sPlot) and literatures. The idea of quantifying VSC based on traditional field survey data can easily be extended, thereby providing ideas for predicting spatial variation in productivity at the global scale.”(Line 49-58)

#5 the hypothesis was presented as results (line 133-135 – verb tense) and, therefore, did not reasonable displaying of the outcomes, either explaining the potential mechanisms such as the shape and direction of changing the resource limiting factors.

Response: In this part of the results, we summarize the previous multiple regression table, because we just want to see whether the change trend is consistent with the hypothesis. We illustrate this and correct the tense in the Results section. We have made the following revisions: “Summary based on multiple regression in Table 2 (original samples) and Table S3 (random samples), when climatic conditions gradually change from favorable to extreme (transition from forest to alpine grassland ecosystem), the effects of resource availability on the variation in VSC were gradually decline from subtropical forests to alpine desert grasslands, and the effects of non-resource limiting factors were become larger (Fig. 5).”(Line 146-150)

#6 Moreover, the comparison between maximum height within latitudes over the globe (other studies – considering other ecosystems and vegetations type) and the TP, I could not follow as the reason why max height was selected was not fully presented. The discussion section follows the same pattern. The first paragraph brings a study using Altimeter data followed by comparisons, so it would be better to state the main findings accordingly to the expectations.

Response: We have made a lot of changes to the comparison between TP height and the world. Specifically, we used a new data set involving more vegetation types and more scientifically credible for comparison. As you emphasized, the plant height value of the previous data set used for comparison was low (the author of the data set also discussed emphasizes this point). We describe it in detail in comment #23.

#7 Some issues with the writing need much improvement, and please see the minor concerns.

Response: Thank you for your suggestions. Full text has been checked.

Minor concerns

#8 Line 31: the definition of VSC is not clearly stated. I would suggest looking at the review paper by Atkins, Fahey, et al., 2018 and other papers Bergen et al., 2009; McElhinny et al., 2005; and, Müllerová et al., 2021.

Response: We have revised and improved the definition of VSC: “Vertical structural complexity (VSC) in plant heights, which is physical niche partitioning in above-ground space, is thought to be an important property for a specific plant community ².”(Line 32-33)

#9 Line 34: Barry et al. 2019 proposes that complementarity and not only complementarity effects (as it is only a quantification of overyielding in mixture relative to monocultures) can happen due to different causes and one of them how species differ in the use of resources. This sentence is not clear stating what Barry et al. 2019 proposes, so I would re-write it in way that likely species that use resources differently may display different functional traits expressed in the VSC.

Response: We have revised this section to make it more precise: “For example, the classical complementary effect ³ proposes that the spatial complementarity due to VSC is a key determinant of overyielding in species-rich communities^{4,5,6}, as more complex vertical structures mean stronger spatial niche partitioning, thereby reducing competition ^{7,8} and leading to the unique occupancy of niche axes such as light ^{1,4}.”(Line 34-38)

#10 Line 37: is highlight the importance of light harvesting and light-use efficiency via VSC, but any radiation measurements were considered in this study, I would give another example.

Response: We have made the following revisions: “as more complex vertical structures mean stronger spatial niche partitioning, thereby reducing competition ^{7,8} and leading to the unique occupancy of niche axes such as light ^{1,4}.”(Line 37-38)

#11 Line 39-40: I would be careful with these statements (even using your own review) as cited before, the VSC have been deeply used even in large-scale remote sensing studies.

Response: After careful consideration, we feel that this statement is not rigorous enough and have deleted it.

#12 Line 43-45: Again ... TLS, GEDI are open-access data available and not coarse-grained atlases.

Response: This part of the statement did lead to ambiguity, and we have corrected the sentence and added a reference: “Despite recent advances in airborne laser scanning-based VSC studies at the global scale, their usefulness is somewhat hampered by the limited number of sample sites ⁹.”(Line 51-52)

#13 Line 49: is there any citation to support this statement?

Response: We have added a reference to this sentence: “PHT reveals VSC spatial variation mainly from the perspective of regional differences in resource availability ⁹.” (Line 60-61)

#14 Line 59: This sentence reads not clear: maybe a suggestion “PTH help to some extent the understanding of VSC spatial variation.”

Response: Thanks for the correction, we have made revisions: “The resource acquisition strategies (resource availability and competition for resources) highlighted by PTH and asymmetric competition help to some extent the understanding of VSC spatial variation^{9,13}.”(Line 70-72)

#15 Line 65: hard time to understand these terms: resource availability and non-resource limiting factors...

Response: Because two reviewers had opposing opinions on using the term "non-resource limiting factors", I have defined "non-resource limiting factors" in this revision where it first appears: “non-resource limiting factors (that is, environmental factors that cause physiological restrictions on plant growth and reproduction without providing resources)”(Line 78-80)

#16 Line 65-66: will gradually become smaller with what? The sentence for me is not complete.

Response: We have made the following revisions: “Therefore, we assumed that as climatic conditions shift from favorable to extreme, the impact of resource availability on spatial variation of VSC will decrease as environmental stress increases, while the effect size of non-resource limiting factors (that is, environmental factors that cause physiological restrictions on plant growth and reproduction without providing resources) such as extreme low temperatures, high ultraviolet radiation, and strong wind will gradually increase (Fig. 1).”(Line 76-81)

#17 Line 67: complete the sentence. “will gradually increase VSC”

Response: What I want to express here is that the impact of non-resource limiting factors on the spatial variation of VSC will become larger. Revisions have been made to eliminate ambiguity. See comment #16.

#18 Line 87-89: why were they selected? not introduced before. Why can variation in height display VSC? Just follow the reasoning such as different patterns of tree height, can provide insights into habitat adaptation, coexistence, evolution, etc.

Response: We have made the following revisions: “These three parameters can fully describe the spatial niche occupancy of community individuals in the vertical dimension.”(Line 102-103)

#19 Line 115: use of abbreviation before the term has been introduced AI was not introduced before in this sequence of the manuscript that I’ve got it.

Response: We have made the following revisions: “Aridity index (AI)”. (Line 129) We also examined the full text for possible similar issues.

#20 Line 131-137: This section is quite short and confusing as it states the hypothesis as the results. I guess the verb tense is not correctly used, but confirming assumptions is poorly explained.

Response: We have made the following revisions: “Summary based on multiple regression in Table 2 (original samples) and Table S3 (random samples), when climatic conditions gradually change from favorable to extreme (transition from forest to alpine grassland ecosystem), the effects of resource availability on the variation in VSC were gradually decline from subtropical forests to alpine desert grasslands, and the effects of non-resource limiting factors were become larger (Fig. 5).”(Line 146-150)

#21 Line 135: Fig 5 and then later comes Fig. 4 (line 140). So, the presentation and discussion of figures are not in sequence.

Response: We carefully examined these types of issues. Figure 4 has appeared in the second part of the results.

#22 Line150: I would start the discussion with a wrap-up of the findings based on what were predicted.

Response: Done. We begin our discussion with the following sentence: “Whether the vegetation height on the Tibetan Plateau is higher or lower than the global level is one of the focuses of this study, because the height of vegetation is closely related to the carbon cycle^{1,24}.”(Line 164-166)

**#23Line 152: which broaföeaf evergreen forest? Can it also be a tropical forest? then 12 meters is too short. Please clearly state the area. a quick research of CHM of broadleaf and evergreen forest mean height - max can vary between 20 - 60 m in temperate zones...
<https://doi.org/10.3390/rs15040975>
<https://doi.org/10.3390/rs15061522>**

Response: In the study by Simard et al., these vegetation types were mainly in the tropical belt. Simard's study also emphasized that their predicted mean values were much lower than those of Lefsky [2010] et al. because they included fewer vegetation types. In our new revised version, we re-contrast the height of forests on the Tibetan Plateau with the study of Lefsky et al., which provides a global range of numerous vegetation types. We have revised the Discussion section as follows: “Lefsky , et al. (2010) used Spaceborne lidar sensors data to map the height of the forest canopy on a global scale. Compared with the global-scale study by Lefsky et al.(2020), we found that the maximum height of the Tibetan Plateau forest (23.45m) is only lower than the global temperate coniferous forest (27.4m), but significantly higher than the global tropical subtropical dry broadleaf forest (16.9 m) and boreal forest (14.5 m) (Fig. S4).”(Line 166-170)

#24 Line 155: I was impressed with 23 meters maximum height measured with a telescopic stick. How could it be done? The methods section did not explain exactly who the stick was used to measure the top canopy of the trees in the forests.

Response: We have provided further supplementary explanations on this part, as follows: “The maximum measurement scale of the telescopic stick is 20m, and the telescopic stick is marked with measurements accurate to centimeters. For those trees with a height greater than 20m, we complete the measurement based on climbing the trunk.” (Line 292-295)

#25 Line 155-156: there was no introduction to why max height was so important to measure. Global average of broadleaf forest in Temperate to arctic zones... please be specific –

Response: We added a sentence at the beginning of the discussion emphasizing why it is important to measure the maximum height of vegetation: “because the height of vegetation is closely related to the carbon cycle ^{1,24}.” (Line 165-166) We also added a description of why we should compare it to the Arctic tundra : “Compared with arctic tundra, which represents an extreme environment similar to the Tibetan Plateau” (Line 175-176)

#26 Line 157: be specific to your area; think about tall trees in the tropics, only precipitation is not the most important factor; light and temperature influence a lot.

Response: As you emphasize that light and temperature are also important factors affecting tree height, we only emphasized precipitation in this study because this study and some similar studies found precipitation to be the strongest predictor of tree height.

#27 Line 160: could compare with other alpine grasslands... European Alps?

Response: This study only compares the alpine grasslands of the Tibetan Plateau with the polar regions and does not involve other alpine areas because the Tibetan Plateau is known as the "Third Pole of the World." We hope to make a comparison from an angle that is of great interest

to everyone, that is, both the Tibetan Plateau and the Polar Regions have extreme climates, and how different the vegetation heights are between the two places.

#28 Line 161- 162: why is it important to compare alpine grasses with arctic tundra dominated by lichens???

Response: See comment 27.

#29 Line 163: the missing year in the citation

Response: Done.

#30 Line 164: 36% lower in maximum height?

Response: To avoid ambiguity we added a description: "in maximum height". (Line 182)

#31 Line 165- 166: global middle latitudes. So, I really do not understand this sentence. What is the relation of the high latitudes' maximum height of TP with the lowest global regions?

Response: We This is because the high latitude regions of the Tibetan Plateau have the lowest vegetation heights on the entire Tibetan Plateau

#32 Line 183: I can not understand this sentence. PTH is not the hypothesis? I think here the predictors' name is missing.

Response: In the revised version we changed "PTH" to "resource availability". (Line 200)

#33 Line 186: how did Tdiurnal affect height max? state the directional relationship that the reader can try to follow your argumentation. Where can I see this? Fig?

Response: We noted tables that would support this idea. (Line 204)

#34 Line 191: Throughout the text: why the variables have first capital letters in the text?

Response: The use of capitalization us to make it more recognizable.

#35 Line 192: why discuss climate change now? Why does not bring it to the point of the hypothesis?

Response: Wind is a non-resource limiting factor in our study. To make it easier for readers to understand, we added parentheses to mark it. (Line 208) We found a strong influence of wind on Height-max in the alpine steppe, which supports our hypothesis, which was emphasized at the beginning of this paragraph. When we talk about climate change, we hope to make predictions about future changes in altitude through our results and previous findings.

#36 Line 196: replace more for different.

Response: Done.

#37 Line 198: Why defense here? It was never introduced before.

Response: We are aware that this term may cause ambiguity. Referring to the introduction of this article, we have revised it as follows: “achieving greater fitness to avoid mortality.”(Line 215-216)

#38 Line 206: why the community will be more uniform if you have facilitation? It could well be a facilitation via complementarity and then they will be able to diversity in structure.

Response: Compared with competition, facilitation can tolerate the overlap in spatial niches between individuals and lead to the homogenization of community structure. Therefore, under extreme climate conditions, we can find that the height distribution of individuals in natural communities is very consistent, because if an individual is higher than the overall height of the community, the probability of being harmed by extreme environmental conditions is greater. We have made the following revisions: “Third, under extreme environmental conditions, competition between plants is weakened, and the facilitation will be enhanced for resistance to adverse environmental conditions; compared with competition, facilitation can tolerate the overlap in spatial niches between individuals and lead to more uniform community structure¹⁷.”(Line 222-225)

#39 Line 213: I would think the opposite: biological diversity promotes the complex vertical structure of plant communities.

Response: We have made the following revisions: “higher and more complex vertical structures can promote the coexistence of species at other trophic levels by providing more ecological niches²⁵.”(Line 235-237)

#40 Line 244: wording Exciting

Response: Thanks for your suggestion, it has been revised. (Line 271)

#41 Line 246- 247: this is the first time limiting factors are defined as they seem to be.

Response: Based on the opinions of the two reviewers, we have defined non-resource limiting factors in the introduction.(Line 78-80) Here, the "limiting factor" is also changed to "non-resource limiting factor" to ensure uniformity throughout the text.(Line 274)

#42 Line 250-251: no temporal change results?

Response: We Because this study surveyed a large number of sample sites, it took three years to complete the field investigation. Each site is not repeatedly sampled across the time gradient.

#43 Line 255: How was max height measured with a stick?

Response: We have made the following revisions: “We determined the tallest plant in each plot by visual inspection, and the height of the tallest plant in each plot was measured with a steel ruler.” (Line 283-285)

#44 Line 257: three plant individuals per species?

Response: For grassland ecosystems, because each plot contains a large number of individuals, we randomly measured the heights of three plant individuals for each species.

#45 Line 256: Missing methods on how height was measured in grasslands.

Response: We have made the following revisions: “we recorded their coverage and then randomly selected three plants of each species to measure their height with a steel ruler” (Line 283-285)

#46 Line 259-260: This sentence is confusing, please consider to rewriting it. For forests, there were 456 plots and shrubs 30 plots. How was the size of all plots? The three plots 20 x 20 means what? for forests or for shrubs and why only 3 randomly?

Response: We rewrote this sentence: “For forest and shrubland ecosystems, three 20 m× 20 m plots were randomly established in each grid, for a total of 486 plots (Forests: 456; shrublands: 30; Table 1).” (Line 288-289) The three plots in each grid refers to many previous ideas for plot layout.

#47 Line 270: Is the height-max averaged per plot?

Response: Height-max is the height of the tallest plant individual in each quadrat. This has been reinforced in the revision to the previous comment.

#48 Line 288: what is MAP? Not defined before? as you did to AI.

Response: MAP has been explained in the study area section, which is the mean annual precipitation. In order to obtain maximum prediction accuracy when making predictions with the machine learning model, we did not delete variables with collinearity. In the subsequent multiple regression analysis, in order to evaluate the relative importance between factors, we screened the variables used for model construction based on the collinearity between factors. MAP has a strong correlation with AI, so only AI was retained in subsequent analyses.

#49 Line 294: Light is super important. But not considered.

Response: The vertical distribution of plant communities is largely to make full use of light resources, but research shows that the photosynthetically active radiation between each point is in a saturated state. It is more affected by the competition for light resources within the local community and the shade tolerance of plants.

#50 Line 296-297: Although the study considers a large area, they are still comparing different ecosystems and measuring at the local level. This argumentation is weak. Second, the argument here can be discussed (different kinds of literature cite other results), but the point is PAR is fundamental for VSC as it is directly related to the plant structure. Leaf angles, arrangement, and distribution across height strata of a plant. So, it is important for resource acquisition such as light, water and biomass production. This is just not fit for this study.

Response: When we obtained climate data, we obtained ultraviolet radiation and photosynthetically active radiation data at the same time, and the correlation between the two is very strong (> 0.95). At the same time, in our results (Table 2), we can also find that ultraviolet radiation has a weak impact on the spatial variation of CSV.

#51 Line 299: This is citing Callaway, which found that in a short experiment, plant species displayed a strong shift from competitive interactions in a relatively good EC to facilitation processes in more stressful EC. So, this sentence is not fully correct with the cited paper.

Response: Thanks for your reminder, we have changed the reference to a more appropriate one.

#52 Line 302: Different types of abbreviations. Please be consistent. Please write within the brackets for minimum temperature (MAT)

Response: Done.

#53 Line 307: sample randomly within an area of 400 m². How exactly was the area chosen?

Response: Because the number of replicates of 15-30 soil samples collected within a 400 m² plot is sufficiently representative, we did not follow any rules during the specific sampling process and it was a completely random sampling.

#54 Line 318: missing year of the citation

Response: Done.

#55 Line 338: First, remind the reader of your assumptions again.

Response: We highlight the assumptions in this section: “(that is, the underlying mechanisms that dominate VSC variation will gradually shift from resource acquisition strategies to conservation priority strategies).” (Line 370-372)

#56 Line 339: typo

Response: Done.

#57 Line 344: it is not after? It is just confusing further. I would move this paragraph before the MSR.

Response: We have revised the beginning of this paragraph as follows: “When conducting MSR analysis, we screened explanatory variables based on the following principles.” (Line 378-379)

Reviewer: 2

#58 Here, in this interesting study, the authors aim to explore how the plant community vertical structural complexity (VSC) vary and how the driving factors are and also establish a VSC map in Tibetan Plateau, China. It is surprising the author to explore these topics by field sampling of so large number of plots, a very difficult work as far as I know. The topic of this study is very clear and important, and the results are convincing based on such great dataset, and the writing is nice. I would not repeat the main results of this paper and give a few of comments after I completely reviewing this paper.

Response: We appreciate your positive comments on this study. We have made revisions to the relevant issues you raised after careful consideration.

#59 L18 and throughout this paper: It is better to use the full name of the Tibetan Plateau rather than the abbreviation of TP.

Response: We have revised the text by replacing "TP" with "Tibetan Plateau" to increase the readability of the article.

#60 L53: I still wonder why the warmer and more humid sites have more plant species with different strategies. This is a generally phenomenon, and there are many mechanisms for why it occurs and I'd like to see your idea on this topic. As you also mentioned, the harsh sites usually accommodate few species, and it is easy to understand this; while how it will be more diverse of the plant community in the agreeable conditions. This is also a important question as when we looking into plant roots and the authors could refer to a paper by Laughlin et al., 2021 NEE.

Response: We have carefully studied this article you recommended, which has expanded our horizons. In this study we argue that in rather large environments, communities are more complex in composition, or have a wider range of functional strategies. Such a statement is based on the classic community construction theoretical framework, that is, the environmental filtering effect of regional geography and the species diversity of local community geography jointly dominate community construction. When environmental conditions change, the intensity of the filtering effect will be higher, and more species in the species pool will be able to survive in the local community. Some scholars describe more energy promoting more diverse communities as “a pie that can be divided into more pieces.”

#61 L56-57" It is true that the big plants can reduce the small plants in a community, for example, and the big plants could be more. Then why such big plants dominated communities can lead to greater VSC? The authors could explain it more clearly.

Response: We have revised this sentence as follows to make the expression more precise: “Competition often results in inconsistent individual sizes and promotes complex vertical structures, because larger individuals gain more resources per unit of biomass and should inhibit the growth of their smaller neighbors”.(Line 66-69)

#62 L60-62: Here I understand the tradeoff refers to different plant communities in different environmental conditions rather than the same environment in different conditions, e.g., the favorable or the extreme conditions. Therefore, I suggest to deleting the "specific" in L60.

Response: Done.

#63 L66: It is cool using the non-resource limiting factors and resource limiting factors, and I like it.

Response: Thank you for your recognition.

#64 L159: add a reference

Response: We added a link to this sentence: “The forests in the southeastern corner of the TP have always held records for the tallest trees in China (<https://new.qq.com/rain/a/20230527A062HA00>).”(Line 175)

#65 L205 “facilitation of common resistance” is confusing, please rephrase.

Response: We have made the following revisions to this part : “the facilitation will be enhanced for resistance to adverse environmental conditions.”(Line 223-224)

#66 L215-216 Here the authors refer to the functional diversity and multifunctionality. Could you please explain these terms clearly and why the VSC can enhance these diversity as from its current form I do not understand why variation of the VSC result in the functional diversity change.

Response: We have made the following revisions to this part : “(iii) complex vertical structures mean communities have diverse physiological traits (such as shade tolerance, crown plasticity, etc.), which will affect the functional diversity of the community and thus the multifunctionality of the entire ecosystem.”(Line 237-239)

#67 L217 Yeah, these maps are very important, and here I’d like to know whether the authors considering the temporal variation of the plant height, variance and evenness into the model as it is possible the communities may not all be in the climax stage of the community succession. This is just a suggestion and I hope the authors could take this point into their discussion.

Response: Thank you for your valuable suggestions, we agree that considering different successional stages during model construction will make the results more reliable. Unfortunately, this study has not yet collected relevant data. We emphasized this point in our discussion. “In addition, plant communities in different successional stages have different resource utilization and acquisition strategies, however, this study failed to consider this during the model construction process due to lack of data⁹.”(Line 249-252)

Thanks again

Yours sincerely,

Nianpeng He

Reviewers' comments:

Reviewer #1 (Remarks to the Author):

Peer reviewing - Second round – Biology Communications COMMSBIO-23-0888-T

Manuscript title: " Vertical structural complexity of plant communities represents the combined effects of resource acquisition and environmental stress on the Tibetan Plateau."

I would like to thank the authors of the current manuscript for the much-improved version of the submitted article. The revisions were extensive and indeed improved the manuscript. I appreciated the deeper explanation of the term "non-resource limiting factor" by stating its proper meaning. Thank you very much for doing it.

However, I still have a few points that might be important for this research.

i) Citation:

In the first revision, I suggested being more careful with statements/argumentation about your methodological approach, considering the advances in this field with LiDAR and terrestrial laser scanning. In the current version, the authors argued: "Despite recent advances in airborne laser scanning-based VSC studies at the global scale, their usefulness is somewhat hampered by the limited number of sample sites 9." However, this citation LaRue et al. 2023 does not mention LiDAR technology in any example. This study (LaRue et al. 2023) used inventory data from different US Forest Services. Hence, I did not understand why this article was chosen for this argumentation. My point here is that there is no problem with using traditional or remote sensing techniques to estimate the vertical structural complexity of the vegetation; all methods are well-established and work well. The issue here is wrongly arguing the advantages of one method over the other and using the wrong citation.

Moreover, the authors compared their traditional inventory data results with a global LiDAR study (Lefsky 2010; Geoph. Res. Letter) in the discussion section. This somehow contradicts the authors' argument above. My suggestion is to minimize this comparison with the advances in mapping global forest canopy height using LiDAR data (such as Potapov et al. 2021 Remote Sensing of Env.; or Simard et al. 2011 J. Geoph. Res.) and focus on what your method found within all the ecosystems in the Tibetan Plateau as this is novel in this study.

(LaRue, E. A. et al. Structural diversity as a reliable and novel predictor for ecosystem productivity. *Front. 432 Ecol. Environ.* 21, 33-39, doi:10.1002/fee.2586 (2023).)

ii) Figure depiction

In the depiction of height-evenness and height-var in Fig. 1, I was confused as I expected an opposite pattern for the two variables. For instance, a plant community with a low height-var likely will present high evenness, meaning that the biomass is more homogeneously distributed along height strata and therefore displays a lower variance in height. I expected a similar depiction in the graphic for low height-var and high height-even. Please check if it is correct according to the author's expectations.

iii) Mechanisms driving VSC effects – non-resource limiting factor.

Although I appreciated the explanation for plant communities such as grassland (tighter packed, clumpier, and therefore ameliorating each other's microclimate – via facilitation), for forest regions, I could not follow how aridity (as the most dominant factor) could influence the homogeneity of canopy structure of the trees. Could the authors add a sentence explaining the mechanism behind these results?

iv) Language:

Line 149 - The verb tense is still not correct. The correct version is "the effects of resource availability on the variation 148 in VSC were gradually declined from subtropical forests to alpine desert grasslands."

Line 150 – Please correct. "effects of non-resource limiting factors have become larger (Fig. 5)."

Point-to-point responses to the comments

(COMMSBIO-23-0888A: Vertical structural complexity of plant communities represents the combined effects of resource acquisition and environmental stress on the Tibetan Plateau)

Dear Reviewers,

Thank you very much for your encouragement and comments which greatly improved the paper. We have revised the manuscript thoroughly. Furthermore, we have invited the native English speaker in the company of editage (www.editage.cn), to polish our English usage.

Here, we provide four files to show how we revised the manuscript: 1) revised main file; 2) revised main file with tracked changes; 3) revised supplementary file; 4) point-to-point responses to reviewer's comments (attached below).

Thank you for considering this paper, which we have revised to be timely and exciting for Communications Biology.

Yours sincerely,

Nianpeng He

Institute of Geographic Sciences and Natural Resources Research, CAS

Datun Road A 11, Chaoyang District, Beijing 100101, P.R. China

E-mail: henp@igsnrr.ac.cn

Tel: 86-10-64889263

Fax: 86-10-64889432

Reviewer: 1

Reviewer's comments:

1 Citation: In the first revision, I suggested being more careful with statements/argumentation about your methodological approach, considering the advances in this field with LiDAR and terrestrial laser scanning. In the current version, the authors argued: “Despite recent advances in airborne laser scanning-based VSC studies at the global scale, their usefulness is somewhat hampered by the limited number of sample sites 9.” However, this citation LaRue et al. 2023 does not mention LiDAR technology in any example. This study (LaRue et al. 2023) used inventory data from different US Forest Services. Hence, I did not understand why this article was chosen for this argumentation. My point here is that there is no problem with using traditional or remote sensing techniques to estimate the vertical structural complexity of the vegetation; all methods are well-established and work well. The issue here is wrongly arguing the advantages of one method over the other and using the wrong citation.

Response: We are very grateful to the reviewer for his valuable suggestions and agree with his view that both field surveys and airborne laser scanning are relatively mature methods for quantifying vertical structural complexity. After careful consideration, we believe that the differences in comparison methods in the introduction do not fit well with the theme of this research, so we have removed this part of the content in the new version so that the logic will be more compact. At the same time, we carefully checked the rigor of literature citations in the new revised version.

#2 Moreover, the authors compared their traditional inventory data results with a global LiDAR study (Lefsky 2010; Geoph. Res. Letter) in the discussion section. This somehow contradicts the authors’ argument above. My suggestion is to minimize this comparison with the advances in mapping global forest canopy height using LiDAR data (such as Potapov et al. 2021 Remote Sensing of Env.; or Simard et al. 2011 J. Geoph. Res.) and focus on what your method found within all the ecosystems in the Tibetan Plateau as this is novel in this study. (LaRue, E. A. et al. Structural diversity as a reliable and novel predictor for ecosystem productivity. Front. 432 Ecol. Environ. 21, 33-39, doi:10.1002/fee.2586 (2023).)

Response: We have removed the comparison with LiDAR-based results in the revised version. At the same time, we have added some new content at the beginning of the discussion to focus the discussion on the core findings of this paper. We've added the following: “Based on large-scale field survey data on the Tibetan Plateau, this study explains the determining mechanism of spatial variation in VSC, and produces 1-km resolution spatial atlas based on machine learning models.”(Line 157-159)

#3 Figure depiction. In the depiction of height-evenness and height-var in Fig. 1, I was confused as I expected an opposite pattern for the two variables. For instance, a plant community with a low height-var likely will present high evenness, meaning that the

biomass is more homogeneously distributed along height strata and therefore displays a lower variance in height. I expected a similar depiction in the graphic for low height-var and high height-even. Please check if it is correct according to the author's expectations.

Response: The findings of this paper are also consistent with the reviewers, that is, these two variables show opposite patterns. Taking three types of alpine grasslands as an example, we found that for Height-var meadow grasslands are larger than steppes grasslands and desert grasslands, while for Height-even it is exactly the opposite. We have modified Figure 1 according to the reviewer's comments to avoid this confusion.

Fig. 1

#4 Mechanisms driving VSC effects – non-resource limiting factor. Although I appreciated the explanation for plant communities such as grassland (tighter packed, clumpier, and therefore ameliorating each other's microclimate – via facilitation), for forest regions, I could not follow how aridity (as the most dominant factor) could influence the homogeneity of canopy structure of the trees. Could the authors add a sentence explaining the mechanism behind these results?

Response: Greater moisture conditions mean more complex forest structure. I think two mechanisms can explain this phenomenon. One is that areas with better moisture conditions have more species, and plant communities under better moisture conditions will grow more leaves and branches, which means stronger competition for resources (especially for light resources within the community). In this case, the forest canopy will show higher plasticity,

allowing plants to occupy more spatial niches and avoid excessive competition. We clarify this in the second paragraph of the introduction to the revised edition. (Line 47-58)

#5 iv) Language: Line 149 - The verb tense is still not correct. The correct version is "the effects of resource availability on the variation 148 in VSC were gradually declined from subtropical forests to alpine desert grasslands." Line 150 – Please correct. "effects of non-resource limiting factors have become larger (Fig. 5)."

Response: We have made the following revisions: "when climatic conditions gradually change from favorable to extreme (transition from forest to alpine grassland ecosystem), the effects of resource availability on the variation in VSC were gradually declined from subtropical forests to alpine desert grasslands, and the effects of non-resource limiting factors have become larger (Fig. 5)." (Line 139-142)

Thanks again

Yours sincerely,

Nianpeng He